# Adversarial representation learning for private speech generation

**David Ericsson** [* 1 2]  **Adam Östberg** [* 1 2]  **Edvin Listo Zec** [2]  **John Martinsson** [2]  **Olof Mogren** [2]

## Abstract

As more data is collected in various settings across organizations, companies, and countries, there has been an increase in the demand of user privacy. Developing privacy preserving methods for data analytics is thus an important area of research. In this work we present a model based on generative adversarial networks (GANs) that learns to obfuscate specific sensitive attributes in speech data. We train a model that learns to hide sensitive information in the data, while preserving the meaning in the utterance. The model is trained in two steps: first to filter sensitive information in the spectrogram domain, and then to generate new and private information independent of the filtered one. The model is based on a U-Net CNN that takes mel-spectrograms as input. A MelGAN is used to invert the spectrograms back to raw audio waveforms. We show that it is possible to hide sensitive information such as gender by generating new data, trained adversarially to maintain utility and realism.

## 1. Introduction

With greater availability of computing power and large datasets, machine learning methods are increasingly being used to gain insights and make decisions based on data. While providing valuable insights, the methods may extract sensitive information which the provider of the data did not intend to disclose. An example of this is digital voice assistants. The user provides commands by speaking, and the speech is recorded through a microphone. A speech processing algorithm infers the spoken contents and executes the commands accordingly. However, it has been shown that such state-of-the-art methods may infer other sensitive attributes as well, such as intention, gender, emotional state, identity and many more (Srivastava et al., 2019). This raises the question of how to learn representations of data to such applications, which are useful for the intended purpose while respecting the privacy of people.

Speakers' identities can often be inferred based on features such as timbre, pitch, and speaker style. *Voice morphing* techniques focus on making it difficult to infer information from these attributes by altering properties such as pitch and intensity. However, this often limit the utility of the signal, by altering intonation or variability. *Voice conversion* approaches instead aim to mimic a specific speaker. In contrast, this paper aims at modelling a distribution over plausible speakers, given the current input signal, and while hiding sensitive attributes.

In this paper, we approach the task of privacy-ensuring voice transformations using an adversarial learning set-up. Generative adversarial networks (GANs) were proposed as tractable generative models (Goodfellow et al., 2014), but have also been adapted to transform data and to provide privacy in the image domain (Huang et al., 2018). We build on these findings, and propose PCMelGAN, a two-step GAN set-up similar to from (Martinsson et al., 2020), that works in the mel-spectrogram domain. The set-up consists of a filter module which removes sensitive information, and a generator module which adds synthetic information in its place. The proposed method can successfully obfuscate sensitive attributes in speech data and generates realistic speech independent of the sensitive input attribute. Our results for censoring the *gender* attribute on the AudioMNIST dataset, demonstrate that the method can maintain a high level of utility, i.e. retain qualities such as intonation and content, while obtaining strong privacy.

In our experiments, the filter module makes it difficult for an adversary to infer the gender of the speaker, and the generator module randomly assigns a synthetic value for the gender attribute which is used when generating the output. However, the proposed method is designed to be able to censor any attribute of a categorical nature. The proposed solution is agnostic to the downstream task, with the objective to make the data as private as possible given a distortion constraint.

---

[*]Equal contribution  [1]Chalmers University of Technology, Gothenburg, Sweden [2]RISE Research Institutes of Sweden. Correspondence to: David Ericsson <daverics@chalmers.se>, Adam Östberg <adamostberg@hotmail.com>, Edvin Listo Zec <edvin.listo.zec@ri.se>.

*Published at the workshop on Self-supervision in Audio and Speech at the $37^{th}$ International Conference on Machine Learning*, Vienna, Austria. Copyright 2020 by the author(s).

## 2. Related work

**Adversarial representation learning.** Research within adversarial learning aims to train two or more models simultaneously with conflicting objective functions. One network which is trained on the main task, and one adversary network that is trained to identify the other network's output. Within the image domain, adversarial learning has had a large success in a wide variety of tasks since the introduction of generative adversarial networks (GANs) (Goodfellow et al., 2014). Examples of such tasks are image-to-image transformations (Isola et al., 2017), and synthesis of facial expressions and human pose (Song et al., 2017; Tang et al., 2019).

Much less work with GANs has been done related to speech and audio. (Pascual et al., 2017) introduce SEGAN (speech enhancement GAN) and thus seem to be the first ones to apply GANs to the task of speech generation and enhancement. The authors train a model end-to-end working on the raw-audio signal directly. (Higuchi et al., 2017; Qin & Jiang, 2018) use adversarial learning to perform speech enhancement for automatic speech recognition (ASR). (Donahue et al., 2018) study the benefit of GAN-based speech enhancement for ASR by extending SEGAN to operate on a time-frequency.

While these works are applying GANs to tackle the challenges within speech, they are limited to a supervised setting. The two most notable works in an unsupervised setting are (Donahue et al., 2019) and (Engel et al., 2019). (Donahue et al., 2019) focus on learning representations in an adversarial manner in order to synthesize audio data both on waveform and spectrogram level, but still show that it is a challenging task, concluding that most perceptually-informed spectrograms are non-invertible.

**Intermediate speech representations.** It is challenging to work on raw waveforms when modeling audio data, due to a high temporal resolution but also a complex relationship between short-term and long-term dependencies. This leads to most work being done on a lower-dimensional representation domain, usually a spectrogram. Two common intermediate speech representations are aligned linguistic features (Oord et al., 2016) and mel-spectrograms (Shen et al., 2018; Gibiansky et al., 2017). The mel scale is a nonlinear frequency scale that is linear in terms of human perception. It has the benefit of emphasizing differences in lower frequencies, which are important to humans. At the same time, it puts less weight on high frequency details, that typically consists of different bursts of noise which are not needed to be as distinguishable. (Engel et al., 2019) trains a GAN to synthesize magnitude-phase spectrograms of note records for different musical instruments. (Kumar et al., 2019) tackle the problem of non-invertible spectrograms by introducing MelGAN: a fully convolutional model designed to invert mel-spectrograms to raw waveforms.

**Adversarial representation learning for privacy.** Adversarial representation learning has also been studied as a method of preserving privacy. More specifically, it has been used with the goal of hiding sensitive attributes under some utility constraint. This work has mainly focused on images and/or videos, and some tasks related to text data (Zhang et al., 2018; Xie et al., 2017; Beutel et al., 2017; Raval et al., 2017).

To our knowledge, (Srivastava et al., 2019) are the first ones to apply privacy related adversarial representation learning to audio data. The authors study the problem of protecting the speaker identity of a person based on an encoded representation of their speech. The encoder is trained for an automatic speech recognition (ASR) task. While the authors manage to hide the speaker identity to some extent, their method also relies on knowing labels for the downstream task.

In the works of (Edwards & Storkey, 2016; Huang et al., 2018) and (Martinsson et al., 2020), the authors apply adversarial representation learning to censor images, without using any downstream task labels.

**Voice conversion.** Voice conversion algorithms aim to learn a function that maps acoustic features from a source-speaker $X$ to a target-speaker $Y$. Some notable works on this involving GANs are (Hsu et al., 2017; Pasini, 2019; Kameoka et al., 2018; Kaneko et al., 2019). Similar to (Kameoka et al., 2018), we do not require any parallel utterances, transcriptions, or time alignment for the speech generation part. (Qian et al., 2018; Aloufi et al., 2019) use voice conversion to study privacy in speech. However, these works differ from our by having a target speaker to which they convert the voice of the input speakers to.

## 3. Problem setting

### 3.1. Private conditional GAN

Private conditional GAN (PCGAN) (Martinsson et al., 2020) is a model that builds upon the generative adversarial privacy (GAP) framework described by (Huang et al., 2017; Huang et al., 2018). Both works study adversarial representation learning for obfuscating sensitive attributes in images. The authors of PCGAN show that by adding a generator to the filter model in the GAP framework strengthens privacy while maintaining utility. The filter network obfuscates the sensitive attribute $s$ in the image, and the objective of the generator is to take the filtered image $x'$ as input and generate a new synthetic instance of the sensitive attribute $s'$ in it, independent of the original $s$.

The filter and the generator networks are trained against their respective discriminators $\mathcal{D}_{\mathcal{F}}$ and $\mathcal{D}_{\mathcal{G}}$ in an adversarial

set up. The discriminator $\mathcal{D}_{\mathcal{F}}$ is trained to predict $s$ in the transformed image $\boldsymbol{x}'$, while the filter $\mathcal{F}$ is trained to transform images that fools the discriminator. The training objective of the filter can be described with the following minimax setup:

$$\min_{\mathcal{F}} \max_{\mathcal{D}_{\mathcal{F}}} \mathbb{E}_{\boldsymbol{x},\boldsymbol{z}_1} \left[ \ell_{\mathcal{F}} \big( \mathcal{D}_{\mathcal{F}}(\mathcal{F}(\boldsymbol{x},\boldsymbol{z}_1), s \big) \right]$$
$$\text{s.t. } \mathbb{E}_{\boldsymbol{x},\boldsymbol{z}_1} \left[ d \left( \mathcal{F}(\boldsymbol{x},\boldsymbol{z}_1), \boldsymbol{x} \right) \right] \leq \varepsilon_1 \tag{1}$$

where $\varepsilon_1 \geq 0$ denotes the allowed distortion in the transformation performed by the filter.

The purpose of the generator $\mathcal{G}$ is to generate a synthetic $s'$, independent of the original $s$. Its discriminator, $\mathcal{D}_{\mathcal{G}}$, takes as input a real image or an image generated by $\mathcal{G}$, and is trained to predict $s$ in the first case, and to predict the "$fake$" in the second, as in the semi-supervised learning setup in (Salimans et al., 2016).

This setup is defined with the following minimax game:

$$\min_{\mathcal{G}} \max_{\mathcal{D}_{\mathcal{G}}} \mathbb{E}_{\boldsymbol{x},s',\boldsymbol{z}_1,\boldsymbol{z}_2} \left[ \ell_{\mathcal{G}} \left( \mathcal{D}_{\mathcal{G}} \left( \mathcal{G} \left( \mathcal{F}\left( \boldsymbol{x},\boldsymbol{z}_1 \right), s', \boldsymbol{z}_2 \right) \right), fake \right) \right]$$
$$+ \mathbb{E}_{\boldsymbol{x}} \left[ \ell_{\mathcal{G}} \left( \mathcal{D}_{\mathcal{G}}(\boldsymbol{x}; \mathcal{D}_{\mathcal{G}}), s \right) \right] \tag{2}$$
$$\text{s.t. } \mathbb{E}_{\boldsymbol{x},s',\boldsymbol{z}_1,\boldsymbol{z}_2} \left[ d \left( \mathcal{G} \left( \mathcal{F}\left( \boldsymbol{x},\boldsymbol{z}_1 \right), s', \boldsymbol{z}_2 \right), \boldsymbol{x} \right) \right] \leq \varepsilon_2$$

where $\varepsilon_2 \geq 0$ is the allowed distortion in the transformation performed by the generator.

### 3.2. MelGAN

MelGAN is a non-autoregressive feed-forward convolutional model which is trained to learn to invert mel-spectrograms to raw waveforms (Kumar et al., 2019). The MelGAN generator consists of a stack of transposed convolutional layers, and the model uses three different discriminators which each operate at different resolutions on the raw audio. The discriminators are trained using a hinge loss version (Lim & Ye, 2017) of the original GAN objective. The generator is trained using the original GAN objective, combined with a *feature matching loss* (Larsen et al., 2015), which minimizes the L1 distance between the discriminator feature maps of real and synthetic audio.

For each layer $i$, let $\mathcal{D}_k^{(i)}(\cdot)$ denote the output from the $k$th discriminator. The feature matching loss is computed as $\mathcal{L}_{\text{FM}}(\mathcal{G},\mathcal{D}_k) = \mathbb{E}_{\boldsymbol{x},\boldsymbol{m}} \left[ \sum_i \frac{1}{N_i} \left\| \mathcal{D}_k^{(i)}(\boldsymbol{x}) - \mathcal{D}_k^{(i)}(\mathcal{G}(\boldsymbol{m})) \right\|_1 \right]$ where $N_i$ is the number of output units in layer $i$, $\boldsymbol{x}$ is the raw audio signal and $\boldsymbol{m}$ is its corresponding mel-spectrogram. The training objectives for the discriminators are then formulated as:

$$\min_{\mathcal{D}_k} \big( \mathbb{E}_{\boldsymbol{x}} \left[ \min\left( 0, 1 - \mathcal{D}_k(\boldsymbol{x}) \right) \right]$$
$$+ \mathbb{E}_{\boldsymbol{m},\boldsymbol{z}} \left[ \min\left( 0, 1 + \mathcal{D}_k(\mathcal{G}(\boldsymbol{m},\boldsymbol{z})) \right) \right] \big). \tag{3}$$

The generator objective is:

$$\min_{\mathcal{G}} \mathbb{E}_{\boldsymbol{m},\boldsymbol{z}} \left[ \sum_{k=1}^{3} -\mathcal{D}_k(\mathcal{G}(\boldsymbol{m},\boldsymbol{z})) \right] + \gamma \sum_{k=1}^{3} \mathcal{L}_{\text{FM}}\left( \mathcal{G},\mathcal{D}_k \right), \tag{4}$$

where $\gamma$ is a hyperparameter controlling the balance between the feature matching and fooling the discriminators.

### 3.3. Our contribution

**Notation.** Let $s \in \{0,1\}$ be a binary sensitive attribute, and $s' \sim \mathcal{U}\{0,1\}$. Let $\boldsymbol{z} \in \mathcal{Z}$ be a noise vector, $\boldsymbol{x} \in \mathcal{X}$ a raw waveform and $\boldsymbol{m} \in \mathcal{M}$ a mel-spectrogram representation of $\boldsymbol{x}$. Let $\mathcal{D}$ be a discriminator, $\mathcal{F}: \mathcal{M} \times \mathcal{Z}_1 \to \mathcal{M}'$ a filter network and $\mathcal{G}: \mathcal{M}' \times \mathcal{Z}_2 \to \mathcal{M}''$ a generator. Let $\mathcal{X}'$ and $\mathcal{X}''$ denote the MelGAN inverted sets of $\mathcal{M}'$ and $\mathcal{M}''$. Each $\boldsymbol{x}$ is paired with a sensitive attribute: $(\boldsymbol{x}_i, s_i)$. Each sample $(\boldsymbol{x}_i, s_i)$ has a corresponding utility attribute $u_i$, only used for evaluation. In our case this is the spoken digit in the recording, i.e. $u_i \in \{0, \dots, 9\}$.

In this work we combine PCGAN and MelGAN to adversarially learn private representations of speech data, and name our model PCMelGAN. The whole pipeline is shown in Figure 1. The speech recording $\boldsymbol{x}$ is mapped to a mel-spectrogram $\boldsymbol{m}$. PCGAN, with its filter and generator modules $\mathcal{F}$ and $\mathcal{G}$, is trained to ensure privacy in the mel-spectrogram. We use a pre-trained MelGAN to invert the mel-spectrogram output of our model $\boldsymbol{m}'' \in \mathcal{M}''$ to a raw waveform $\boldsymbol{x} \in \mathcal{X}''$.

We implement $\mathcal{F}$ and $\mathcal{G}$ using a U-Net architecture similar to (Martinsson et al., 2020). For $\mathcal{D}_{\mathcal{F}}$ and $\mathcal{D}_{\mathcal{G}}$ we use the AlexNet architecture (Krizhevsky et al., 2012) as used in (Becker et al., 2018) for gender classification in the spectrogram domain. We use categorical cross entropy as loss functions denoted by $\ell_{\mathcal{F}}$ and $\ell_{\mathcal{G}}$. The L1-norm is used as the distortion measure $d$. The constrained optimization problem is reformulated as an unconstrained one by relaxing it using the quadratic penalty method (Nocedal & Wright, 2006). The distortion constraint is denoted by $\varepsilon$ and the penalty parameter by $\lambda$. The parameters are updated using Adam (Kingma & Ba, 2014).

As a baseline comparison, we use PCMelGAN where the generator module is excluded. Thus we can directly see how much the generator module adds to the privacy task.

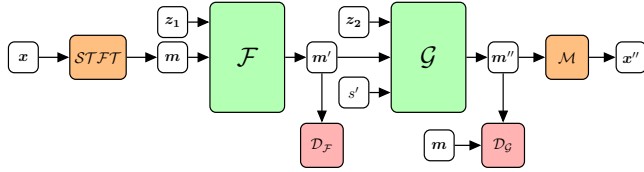

*Figure 1.* Schematic diagram of our model: PCMelGAN.

# 4. Experiments

## 4.1. Data

We use the AudioMNIST dataset to conduct our experiments (Becker et al., 2018). AudioMNIST consists of 30,000 audio recordings of approximately 9.5 hours of spoken digits (0-9) in English. Each digit it repeated 50 times for each of the 60 different speakers. The audio files have a sampling frequency of 48kHz and are saved in a 16 bit integer format. The audio recordings are also labeled with information such as age, gender, origin and accent of all speakers were collected.

In this paper, we use 10,000 samples as a training set and 2,000 samples as a test set. For the training set, we randomly sample speakers such that it consists of 10 female and 10 male speakers. Similarly, the test set consists of 2 female and 2 male speakers. We downsample the recordings to 8 kHz and use zero padding to get an equal length of 8192 for each recording.

## 4.2. Data-driven implementation

To encourage reproducibility, we make our code publicly available [1]. The model is trained end-to-end, with the hyperparameters $\eta_{\mathcal{D}_{\mathcal{F}}}, \eta_{\mathcal{D}_{\mathcal{G}}} = 0.0004$, $\eta_{\mathcal{F}}, \eta_{\mathcal{G}} = 0.0004$, $\lambda = 10^2$, $\varepsilon \in \{0.005, 0.01, 0.05, 0.1\}$ and $(\beta_1, \beta_2) = (0.5, 0.9)$. During training, $m$ is computed using the short-time Fourier transform with a window size of 1024, a hop length of 256 and 80 mel bins. We normalize and clip the spectrograms to $[-1, 1]$ as in (Donahue et al., 2019), with the exception that the normalization is performed on the whole spectrogram as opposed to for each frequency bin.

## 4.3. Evaluation

For each configuration of hyperparameters, we train the model using five different random seeds for 1000 epochs on a NVIDIA V100 GPU. We evaluate the experiments both in the spectrogram and in the raw waveform domain. In each domain, we train digit and gender classifiers on the corresponding training sets, $\mathcal{X}_{train}$ and $\mathcal{M}_{train}$. The classifiers that predict gender are used as a privacy measure, and the classifiers that predict spoken digits are used as a utility measure. We evaluate the fixed classifiers on $\mathcal{M}'_{test}$ and $\mathcal{M}''_{test}$, to directly compare the added benefit by a generator module on-top of the filter.

We also measure the quality of the generated audio using Fréchet Inception Distance (FID) (Heusel et al., 2017). FID is frequently used to measure the quality of GAN-generated images. Since we are interested in measuring generated audio quality, we replace the commonly used Inception v3 network with an AudioNet (Becker et al., 2018) digit

---

[1] https://github.com/daverics/pcmelgan

classifier using the features from the last convolutional layer.

# 5. Results

**Quantitative results.** In Table 1 the mean accuracy and standard deviation of the fixed classifiers on the test set is shown over five runs in the spectrogram and audio domain, respectively. Privacy is measured by the accuracy of the fixed classifier predicting the original gender $s_i$, where an accuracy close to $50\%$ corresponds to more privacy. Utility is measured by the accuracy of the fixed classifier predicting the digit $u_i$, where a higher accuracy corresponds to greater utility.

*Table 1.* The spectrogram classifiers' mean accuracy and standard deviation on the test sets $\mathcal{M}'_{test}$ and $\mathcal{M}''_{test}$ (top) and on $\mathcal{X}'_{test}$ and $\mathcal{X}''_{test}$ (bottom) for varying values of $\varepsilon$. For privacy (gender) an accuracy close to $50\%$ is better. For utility (digit), a higher accuracy is better.

| Dist. | Privacy | | Utility | |
|---|---|---|---|---|
| $\varepsilon$ | Baseline | PCMelGAN | Baseline | PCMelGAN |
| 0.005 | $49.9 \pm 2.2$ | $48.7 \pm 2.4$ | $84.1 \pm 2.8$ | $81.1 \pm 3.7$ |
| 0.01 | $55.0 \pm 4.7$ | $50.9 \pm 1.4$ | $79.9 \pm 4.3$ | $78.8 \pm 7.8$ |
| 0.05 | $61.3 \pm 10.2$ | $51.0 \pm 0.7$ | $80.9 \pm 8.2$ | $54.7 \pm 23.8$ |
| 0.1 | $48.9 \pm 1.0$ | $49.8 \pm 0.5$ | $29.1 \pm 7.5$ | $15.1 \pm 5.4$ |
| 0.005 | $52.2 \pm 3.6$ | $49.1 \pm 1.6$ | $36.8 \pm 4.0$ | $49.4 \pm 9.8$ |
| 0.01 | $53.2 \pm 3.2$ | $51.3 \pm 1.6$ | $34.3 \pm 8.5$ | $49.2 \pm 8.6$ |
| 0.05 | $61.5 \pm 8.1$ | $51.2 \pm 0.7$ | $28.0 \pm 15.8$ | $31.3 \pm 10.3$ |
| 0.1 | $51.0 \pm 1.3$ | $49.6 \pm 0.4$ | $11.4 \pm 1.7$ | $15.8 \pm 2.3$ |

In Table 2, FID scores are shown for our model working in the audio domain. In figure 3, a recording of a woman saying "zero" is shown, together with the baseline (filter) and PCMelGAN generating a male and a female spectrogram.

*Table 2.* The mean FID-score and standard deviation of the test sets $\mathcal{X}'_{test}$ and $\mathcal{X}''_{test}$ for different $\varepsilon$. A lower value corresponds to more realistic audio.

| Dist. | FID Audio | |
|---|---|---|
| $\varepsilon$ | Baseline | PCMelgan |
| 0.005 | $20.17 \pm 4.04$ | $\mathbf{10.12 \pm 3.15}$ |
| 0.01 | $27.27 \pm 4.50$ | $\mathbf{10.02 \pm 2.27}$ |
| 0.05 | $29.59 \pm 5.77$ | $\mathbf{20.22 \pm 4.87}$ |
| 0.1 | $41.50 \pm 3.49$ | $\mathbf{22.32 \pm 5.20}$ |

**Qualitative results.** We provide samples from the AudioMNIST test set that were transformed by our model [2]. The shared folder contains original sound clips and their corresponding transformed versions.

---

[2] https://www.dropbox.com/sh/oangx84ibhzodhs/AAAfG-PBW4Ne8KwdipAmKFy1a?dl=0

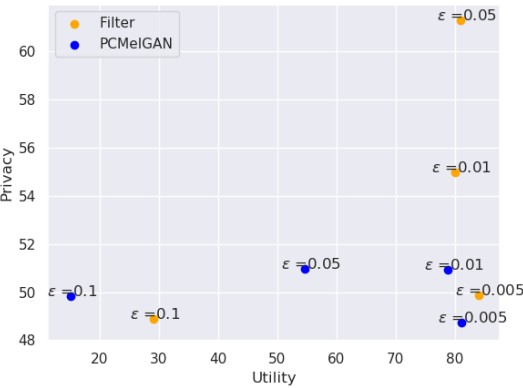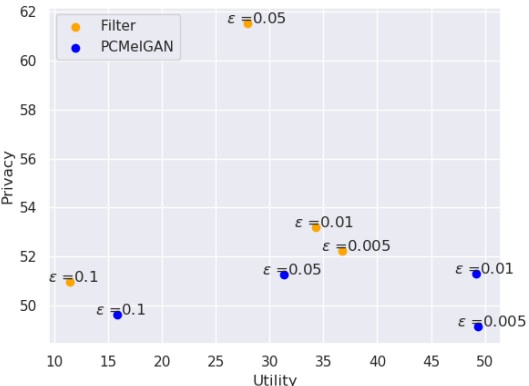

*Figure 2.* Privacy vs utility trade-off for the baseline and PCMelGAN for varying $\varepsilon$. Orange and blue points correspond to evaluating the fixed classifiers for digits and gender on the spectrogram datasets $\mathcal{M}'_{test}$ and $\mathcal{M}''_{test}$ (left), and raw waveform datasets $\mathcal{X}'_{test}$ and $\mathcal{X}''_{test}$ (right). Lower right corner is better.

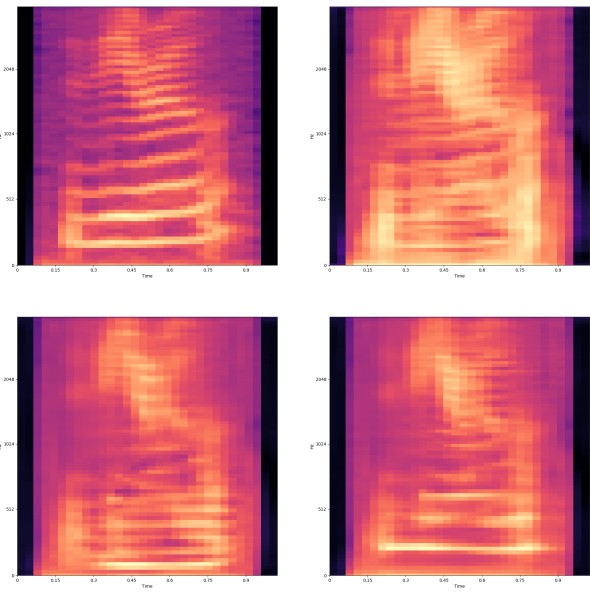

*Figure 3.* Spectrograms of saying "zero". The original recording of a female (top left), transformed ones from the baseline (top right), and our model of a sampled male (bottom left) and a sampled female (bottom right).

## 6. Discussion

Table 1 (top) and Figure 2 (left) demonstrate that the proposed method achieves strong privacy while working on the mel-spectrogram domain, and retains a strong utility preservation. We notice in Table 1 (bottom left) and in Figure 2 (right) that the proposed method is able to provide privacy in the audio domain, but to a loss of utility. However, when comparing to the baseline, we see that generating a synthetic

$s$ both increases utility and ensures privacy. In the spectrogram domain, the filter model seems to be enough to obtain both privacy and utility. In both the spectrogram domain and the audio domain, the proposed approach achieves high privacy. We assume that the privacy will suffer from having a stricter distortion budget $\varepsilon$, but this was not observed in the experiments. While a quick sanity check with $\varepsilon = 10^{-5}$ resulted in the model learning the identity map (with no additional privacy), more experiments need to be carried out to detect when privacy starts to deteriorate with lower $\varepsilon$. It is worth noting that for some $\varepsilon$ we have a large standard deviation. We hypothesize that this could be improved by using more diverse data, and future work should include evaluating the proposed method on longer sentences.

In Table 2 we noticed that our model obtains substantially better FID scores than the baseline in the audio domain. We conclude that adding the synthetic sample of the sensitive attribute improves the realism and fidelity of the speech signal. We observe this also from listening to the generated sounds (see *qualitative results* above).

## 7. Conclusions

In this work we have proposed an adversarially trained model that learns to make speech data private. We do this by first filtering a sensitive attribute, and then generating a new, independent sensitive attribute. We formulate this as an unconstrained optimization problem with a distortion budget. This is done in the spectrogram domain, and we use a pretrained MelGAN to invert the generated mel-spectrogram back to a raw waveform. We compare our model with the baseline of just censoring the attribute, and show that we gain both privacy and utility by generating a new sensitive attribute in the audio domain.

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

# Supplementary

---

**Algorithm 1** PCMelGAN

---

**Input:** dataset $\mathcal{X}_{train}$, learning rate $\eta$, penalty $\lambda$, distortion constant $\varepsilon$

**repeat**

Draw $n$ samples uniformly at random from the dataset
$(x_1, s_1), \ldots, (x_n, s_n) \sim \mathcal{X}_{train}$
Compute mel-spectrogram and normalize
$\boldsymbol{m}_i = \mathcal{STFT}(\boldsymbol{x}_i) \ \forall i = 1, \ldots, n$
Draw $n$ samples from the noise distribution
$\boldsymbol{z}_1^{(1)}, \ldots, \boldsymbol{z}_n^{(1)} \sim \mathcal{N}(0, 1)$
$\boldsymbol{z}_1^{(2)}, \ldots, \boldsymbol{z}_n^{(2)} \sim \mathcal{N}(0, 1)$
Draw $n$ samples from the synthetic distribution
$s_1', \ldots, s_n' \sim \mathcal{U}\{0, 1\}$
Compute the censored and synthetic data
$\boldsymbol{m}_i' = \mathcal{F}(\boldsymbol{m}_i, \boldsymbol{z}_i^{(1)}; \boldsymbol{\theta}_{\mathcal{F}}) \ \forall i = 1, \ldots, n$
$\boldsymbol{m}_i'' = \mathcal{G}(\boldsymbol{m}_i', s_i', \boldsymbol{z}_i^{(2)}; \boldsymbol{\theta}_{\mathcal{G}}) \ \forall i = 1, \ldots, n$
Compute filter and generator loss

$$\mathcal{L}_{\mathcal{F}}(\boldsymbol{\theta}_{\mathcal{F}}) = -\frac{1}{n} \sum_{i=1}^{n} \ell(\mathcal{D}_{\mathcal{F}}(\boldsymbol{m}_i'; \boldsymbol{\theta}_{\mathcal{D}_{\mathcal{F}}}), s_i)$$

$$+ \lambda \max(\frac{1}{n} \sum_{i=1}^{n} d(\boldsymbol{m}_i', \boldsymbol{m}_i) - \varepsilon, 0)^2$$

$$\mathcal{L}_{\mathcal{G}}(\boldsymbol{\theta}_{\mathcal{G}}) = \frac{1}{n} \sum_{i=1}^{n} \ell(\mathcal{D}_{\mathcal{G}}(\boldsymbol{m}_i''; \boldsymbol{\theta}_{\mathcal{D}_{\mathcal{G}}}), s_i)$$

$$+ \lambda \max(\frac{1}{n} \sum_{i=1}^{n} d(\boldsymbol{m}_i'', \boldsymbol{m}_i) - \varepsilon, 0)^2$$

Update filter and generator parameters
$\boldsymbol{\theta}_{\mathcal{F}} \leftarrow \text{Adam}(\boldsymbol{\theta}_{\mathcal{F}}; \eta_{\mathcal{F}}, \beta_1, \beta_2)$
$\boldsymbol{\theta}_{\mathcal{G}} \leftarrow \text{Adam}(\boldsymbol{\theta}_{\mathcal{G}}; \eta_{\mathcal{G}}, \beta_1, \beta_2)$
Compute discriminator losses
$\mathcal{L}_{\mathcal{D}_{\mathcal{F}}}(\boldsymbol{\theta}_{\mathcal{D}_{\mathcal{F}}}) = \frac{1}{n} \sum_{i=1}^{n} \ell(\mathcal{D}_{\mathcal{F}}(\boldsymbol{m}_i'; \boldsymbol{\theta}_{\mathcal{D}_{\mathcal{F}}}), s_i)$
$\mathcal{L}_{\mathcal{D}_{\mathcal{G}}}(\boldsymbol{\theta}_{\mathcal{D}_{\mathcal{G}}}) = \frac{1}{n} \sum_{i=1}^{n} \ell(\mathcal{D}_{\mathcal{G}}(\boldsymbol{m}_i''; \boldsymbol{\theta}_{\mathcal{D}_{\mathcal{G}}}), fake)$

$$+ \frac{1}{n} \sum_{i=1}^{n} \ell(\mathcal{D}_{\mathcal{G}}(\boldsymbol{m}_i; \boldsymbol{\theta}_{\mathcal{D}_{\mathcal{G}}}), s_i)$$

Update discriminator parameters
$\boldsymbol{\theta}_{\mathcal{D}} \leftarrow \text{Adam}(\boldsymbol{\theta}_{\mathcal{D}}; \eta_{\mathcal{D}}, \beta_1, \beta_2)$
**until** termination criterion is met

---