# OpenReview forum: "Adversarial representation learning for private speech generation"
_ICML.cc/2020/Workshop/SAS — SAS 2020_

### Official Review · AnonReviewer1 · 2020-06-25

**Rating:** 7
**Confidence:** 3

**Review:**

1. The paper proposes a method to use adversarial learning for private speech generation. The method consists of 2 classifiers, one to classify sensitive information (gender in this case) and another one for utility (recognition  audio MNIST digits in this case).

2. The idea seems viable, it seems to improve privacy (table 2). Although, there seems to be a tradeoff, it seems that the model performs worse (sometimes significantly worse) in terms of utility (compare to baseline) methods in order to protect privacy.

3. The ablation study (shown in Fig 2) to study the tradeoff betwen utility and privacy is nice.

4. Figure 1 can benefit from a more detailed caption.

5. Overall, the method seems to be promising, although the dataset the model was evaluated on is rather limited. It is significantly easier to "filter" out sensitive information in a word (MNIST digit) rather than a sentence. Im curious to see how the model performs on large speech datasets.

---

### Official Review · AnonReviewer3 · 2020-06-29
**Interesting approach, not sure about the results**

**Confidence:** 3
**Rating:** 6

**Review:**

This paper proposes to combine PCGAN with MELGAN, to construct a generative model pipeline that would preserve the privacy of a sensitive attribute of an audio signal while still retaining some utility such as keeping the data classifiable. The example experiment that has been used in the paper is digit MNIST, where the goal is to keep the gender of the speaker unclassifiable, and at the same to keep the ability to discern which digit the utterance corresponds to.

The paper compares their approach with a simpler version of their approach where they do not include the generative modeling block. The results seem to suggest that their approach is able to increase the performance when compared to the baseline they are considering.

I am wondering however how would the performance compare against simple DSP based approaches where we apply low pass filtering to the data, and then train a generative model on the low pass filtered data. (Or more involved DSP approaches) I urge the authors to add at least one, preferably two DSP based baselines to show that their approach is able to bring something to the table.

For instance, this article suggests that high frequency information is important for gender classification:
https://link.springer.com/article/10.3758/s13414-015-0945-y
Overall, I encourage you to compare your algorithm with DSP baselines.

---

### Official Review · AnonReviewer2 · 2020-06-30
**Privacy in speech , applied to generation**

**Confidence:** 4
**Rating:** 6

**Review:**

Privacy in speech processing is a critical issue, both now and in the future. Because this paper sets out to do private speech generation, and manages to do so to some extent, I argue for acceptance despite the following critiques.

Given the stated goal of privacy preserving - isn't operating at either a higher level (raw text with controllable "synthesis attributes", which are varied each generation to obscure the target attribute), or a lower one (per audio frame, distort and remove pieces of the audio which are not necessary to the end task, such as speech recognition) more directly applicable to the stated goal of "We train a model that learns to hide sensitive information in the data, while preserving the meaning in the utterance".

I do not find FID an adequate measure for this task - privacy preservation according to the specified attribute could be tested directly against a high quality speaker or vocal tract length recognizer being poor, alongside checking whether the resulting audio is still equivalent to the original audio put through a high quality speech recognizer or ASR system.

Listening to the samples, I struggled to hear anything at all in a number of the sampled outputs, and many of the original audio files are so short that it is difficult to have context at all about the attribute that will be "masked". AudioMNIST is such a small dataset that it barely serves as a proof of concept here.

Generally I like the concept of this work, and the effort in tuning and gathering results tables is thorough. But the metrics used here do not seem sufficient for either measuring speech privacy preservation, or quality of the resulting audio. Further exploration along these lines should include directly automated recognition error rates, as well as human quality evaluations. However, achieving any amount of results in such a difficult application area is commendable, and I hope the authors will work further on this area.

---

### Decision · Program_Chairs · 2020-07-01

**Decision:**

Accept

**Comment:**

Dear author(s),

Thank you very much for your submission at the ICML2020@SaS workshop (https://icml-sas.gitlab.io/). Based on the scores assigned by the reviewers, we are happy to notify you that your paper was accepted for the workshop.

Please, address the comments of the reviewers and submit the camera-ready version by July 8. We ask the authors to record a 15min video for your talk. At the workshop, we will have the pre-recorded video as well as a live QA session. It is important to keep this time limit, otherwise, your talk will be automatically cut. The deadline for uploading the video is July 8. The detailed instructions for uploading will follow.

Feel free to contact us for any questions!

Best,

The ICML20@SaS organizers:
Mirco Ravanelli
Titouan Parcollet
Dmitriy Serdyuk
Devon Hjelm
Bhuvana Ramabhadran